# LatentBit: Discrete Visual Tokenization with Preserved Continuous Structure

## Abstract

Image and video autoregressive generative models are limited by their reliance on the language-based framework, and converging evidence points to their discrete representation as the bottleneck. Recent work addresses this bottleneck by constraining representational capacity via latent binarization and by scaling codebooks, yielding measurable generation gains. However, binarizing the codes destroys metric structure, coarsens the latent manifold, and degrades reconstruction under the same token budget. We propose an efficient tokenizer that induces a continuous latent manifold on par with continuous representations, without additional GAN refinements or iterative sampling strategies. Our tokenizer learns a discrete vocabulary aligned to a frozen continuous latent geometry, preserving metric structure and delivering competitive reconstruction quality with a scalable codebook. While naively scaling the codebook increases compute and memory demand, we overcome this limitation by decomposing tokens into bits. On top of this, we train a masked-language-model (MLM) generator with bit-wise prediction, and find that the bit-wise strategy yields better likelihood and faster convergence against alternative subgrouping schemes. This work substantially narrows the performance gap between discrete and continuous representations, bringing discrete approaches close to parity with continuous variants in both reconstruction and generation quality.

## 1 Introduction

Motivated by the remarkable breakthroughs of autoregressive generation in natural language processing, the computer vision community has devoted significant effort to adapting these language style frameworks to image and video generation (Yu et al., 2023c; Kondratyuk et al., 2024; Chang et al., 2022; Yu et al., 2023a). Natural language possesses an inherently discrete structure, where each unit (e.g. a single token) can be represented as categorical latents. This discrete nature provides a significant advantage: it yields a well-behaved cross-entropy objective and enables exact likelihood computation with calibrated sampling.

Visual data is fundamentally different in that pixels are intended to represent a continuous spectrum or scene, creating an inherent mismatch with discrete language frameworks. To address this challenge, the computer vision community has pursued two primary approaches: employing continuous generation frameworks, such as diffusion models (Ho et al., 2020; Dhariwal & Nichol, 2021), that naturally handle continuous pixel values, or developing methods (Van Den Oord et al., 2017; Esser et al., 2021) to discretize visual data into token-like representations to leverage existing discrete generation frameworks.

A critical limitation of the discretization approach is that converting continuous visual data to discrete tokens inevitably introduces information loss. Vector quantization (VQ) has become a popular tokenization method, which maps visual features to discrete indices in a learned codebook. Unfortunately, VQ-based approaches exhibit poor scaling behavior due to training instabilities caused by the quantization process and the limited representational capacity of fixed-size codebooks. Even when codebooks are scaled up, they suffer from under-utilization, a phenomenon known as codebook collapse, where only a small subset of codes are actively used during training. Addressing this issue requires nontrivial solutions such as probabilistic code reassignment or carefully pre-seeded code initialization (Yu et al., 2023b).

Lookup-free quantizers (LFQ) (Yu et al., 2023b; Zhao et al., 2024; Mentzer et al., 2023; Weber et al., 2024) were proposed to address these issues by constraining code capacity via sign-based binarization of the latent dimensions and then hashing the resulting bit vectors. While this design improves scaling by limiting per-code capacity, it amplifies the discontinuity of the non-differentiable quantization step, making gradient-based optimization more challenging. Moreover, the extreme binarization collapses magnitudes and destroys metric structure, producing coarse latents whose reconstructions lag behind continuous VAEs. The practical consequence is that decoder capacity and computational resources are redirected toward repairing quantization-induced artifacts, often requiring larger decoders (Xiong et al., 2025) or sampling the lost information via iterative diffusion heads (Sargent et al., 2025; Birodkar et al., 2024).

Continuous latent generators (e.g., diffusion in VAE space) achieve high perceptual quality but do not provide tractable token-level likelihoods or truly parallel decoding, and make token-granular editing and retrieval less natural. Discrete modeling exposes a language-like interface with exact token probabilities, calibrated scoring, and compositional conditioning. In this work, our goal is to bridge discrete and continuous representations, keeping the advantages of discrete tokens without sacrificing the fidelity and geometry of the continuous latent space. We achieve this with an improved discrete tokenization approach that leverages pretrained VAEs to significantly reduce the gap between discrete tokenizers and continuous VAEs. Concretely, we insert additional trainable transformer layers and a quantization layer within the bottleneck of a pre-trained continuous VAE. This simple yet effective approach dramatically reduces the performance gap between discrete and continuous tokenizers while maintaining the advantages of discrete representations.

The benefits of our approach are twofold: (i) by building on continuous pretrained models, the tokenizer allocates its quantization budget to reconstruct the VAE's latent geometry, maximizing codebook utilization and representational fidelity; (ii) the training is stable and fast to converge without auxiliary objectives (e.g., GAN refinement), simplifying the pipeline while achieving state-of-the-art reconstruction quality. Empirically, the resulting discrete tokens close most of the reconstruction gap to the underlying continuous VAE while preserving single-pass decoding and compatibility with language-style generators.

Leveraging this tokenizer, we introduce Bitwise MaskGIT, a novel MaskGIT-style generation framework that operates with binary codebooks. It delivers substantial improvements over the original MaskGIT approach by casting token prediction into bit-planes and replacing single-index predictions with tractable binary decisions.

In summary, the **contributions** of this work include:

1. **Discrete tokens, continuous fidelity.** We show that a learned *discrete* tokenizer on top of frozen VAE bottleneck can *recover the reconstruction quality of the underlying continuous VAE*, enabling language-style modeling in token space without sacrificing perceptual fidelity, effectively bridging a previously open gap between discrete generative models and continuous VAEs. In particular, our tokenizer achieves state-of-the-art performance in both image and video reconstruction and generation tasks.

2. **Efficient ViT tokenizer for images & video.** Building on our first contribution, we introduce a unified, efficient ViT-based tokenizer that operates on frozen continuous VAE latents. We benchmark against previously published baselines and provide extensive ablations showing favorable rate–distortion and scaling behavior.

3. **Bitwise MaskGIT for large-vocabulary token generation.** We cast token prediction into $D = \lceil \log_2 K \rceil$ bit-planes and train a MaskGIT-style token predictor at the bit level, replacing single-index $K$-way predictions with binary decisions. This yields tractable training and inference in large discrete spaces. We evaluate the quality of MaskGIT bitwise generation in spatial and spatio-temporal domains.

## 2 METHOD

We present a binary video tokenizer that eliminates codebook overhead while maintaining reconstruction quality competitive with continuous representations. Our key insight is that learned transforms can map VAE latents to a binary space where simple sign quantization suffices, avoiding the memory and optimization challenges of large codebooks. From a rate-distortion perspective, at a

Figure 1: Overview of the proposed tokenizer with a frozen VAE encoder–decoder and the learnable ViT tokenizer at the bottleneck. The VAE encodes video to $\mathbf{z}$, where the latent is processed by the ViT tokenizer. At the innermost bottleneck Q quantizes latents, to produce binary codes. A symmetric path is subsequently set to decode the frames to pixel space.

fixed bitrate, continuous latents set a performance upper bound, while discretization introduces additional constraints. We therefore study how far a binary, lookup-free quantizer (LFQ) can approach this bound when paired with a lightweight learned transform.

## 2.1 FROM CONTINUOUS LATENTS TO BINARY CODES

We target high-fidelity image and video reconstruction with a discrete bottleneck. Let $\mathbf{x} \in \mathbb{R}^{T \times 3 \times H \times W}$ denote an input video with $T$ frames at resolution $H \times W$. Images are treated as single frame videos in the tokenizer and T is set to 1. We use $\mathbf{z}$ for continuous latents, $\mathbf{z_{ViT}}$ for pre-quantization embeddings, and $\mathbf{b}$ for binary codes. Spatial and temporal stride factors are denoted $(s_h, s_w, s_t)$.

We employ a pretrained VAE (specifically WAN 2.1 (Wan et al., 2025)) with encoder and decoder, kept frozen during training (denoted $E_{CNN}$ and $G_{CNN}$ in Figure 1). This choice enables fair comparison with continuous baselines and reduces computational requirements by approximately $3\times$ compared to end-to-end training. The encoder maps an input video into a compressed latent representation $\mathbf{z} \in \mathbb{R}^{T_z \times C_z \times H_z \times W_z}$, where $T_z = T/s_t$, $H_z = H/s_h$, $W_z = W/s_w$ with typical strides $(s_t, s_h, s_w) = (4, 8, 8)$ and latent channels $C_z = 16$.

Our tokenizer is based on a ViT architecture placed at the bottleneck of the frozen VAE, such that it observes the latent tensor $\mathbf{z}$ and produces a compact sequence of discrete tokens. In particular, we partition $\mathbf{z}$ into non-overlapping 3D patches of size $(k_t, k_h, k_w)$, treating boundary conditions with zero padding to maintain temporal coherence. This yields $N = \lceil T_z/k_t \rceil \cdot \lceil H_z/k_h \rceil \cdot \lceil W_z/k_w \rceil$ patches, each flattened to dimension $d_{patch} = C_z k_t k_h k_w$. Let $\mathcal{P} : \mathbb{R}^{T_z \times C_z \times H_z \times W_z} \to \mathbb{R}^{N \times D_{patch}}$ denote the patchification operator and $\mathcal{U}$ its inverse. The patchified latent is then processed by the ViT encoder $E_{ViT}$ to produce our innermost bottleneck $\mathbf{z_{ViT}} \in \mathbb{R}^{N \times D}$,

$$\mathbf{z_{ViT}} = E_{ViT}(\mathcal{P}(\mathbf{z})). \tag{1}$$

While our bottleneck is quantizer-agnostic, in practice we instantiate it with a lookup-free binary quantizer (LFQ) for its codebook-free scalability and low latency: LFQ removes table lookups and memory-bound codebooks, keeping complexity linear in the bottleneck width. This choice is especially attractive at video scales, yet remains compatible with our interface and can be swapped for alternative quantizers at matched bitrate. In particular, our binary codes $\mathbf{b}$ are obtained as $Q(\mathbf{Z_{ViT}}) = \text{sign}(\mathbf{Z_{ViT}})$. A symmetric transformer decoder $G_{ViT}$ maps binary codes back to the patch space, which we then unpatchify to get the reconstructed continuous latent,

$$\hat{\mathbf{z}} = \mathcal{U}(G_{ViT}(\mathbf{b})). \tag{2}$$

Crucially, we reconstruct in the continuous latent space ($\hat{\mathbf{z}}$) before VAE decoding, which will generate the final frames.

The tokenizer is optimized under the standard evidence lower bound omitting the KL regularization term. This objective is enforced through a pixel reconstruction term $\mathcal{L}_r$, and a perceptual loss obtained from VGG network $\mathcal{L}_p$. We observe a natural code utilization behaviour in the models with smaller number of learnable parameters, hence we omit the entropy objective adopted in prior works Yu et al. (2023b); Zhao et al. (2024). Furthermore, we find the original commitment loss (Esser et al., 2021) yields consistent gains, so we retain it in our objective.

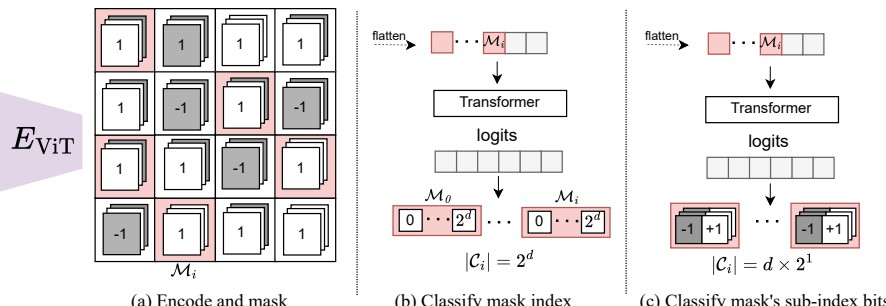

(a) Encode and mask      (b) Classify mask index      (c) Classify mask's sub-index bits

Figure 2: Proposed bitwise MaskGIT framework where (a) We encode the sequence and mask selected tokens (shown in red). (b) Index-based methods predict a single codebook index in $[0, 2^D - 1]$. (c) In contrast bitwise MaskGIT predicts D binary bits for each masked token.

## 2.2 GENERATION

Directly generating tokens over the tokenizer's full codebook is memory-prohibitive. With a 32-bit latent index from $E_{\text{ViT}}$, the discrete space spans $\approx 4B$ classes, each requiring an embedding and a dense connection to the output head. Prior work mitigates this by partitioning the index into sub-groups (Yu et al., 2023b; Zhao et al., 2024); in the limit, one predicts individual bits, as in autoregressive next-scale prediction (Han et al., 2025). We extend this bitwise factorization to MaskGIT-style parallel generation and refer to it as maximal subgrouping. In contrast to bit-masking (Weber et al., 2024), our approach keeps the MaskGIT token masking intact and replaces the token head with a bitwise classifier that composes the final index. Let $\mathbf{b} = (\mathbf{b}_1, \ldots, \mathbf{b}_N)$ denote the bits within the token sequence emitted by the tokenizer, where each $\mathbf{b}_i$ is a binary code. We define a mask as a subset $\mathcal{M} \subseteq \{1, \ldots, N\}$ indicating masked positions, over the sequence and use a learned mask embedding, $\mathbf{e}_{\text{mask}}$, to construct the masked sequence (see Figure 2(a)):

$$\tilde{\mathbf{b}}_i = \begin{cases} \mathbf{e}_{\text{mask}}, & i \in \mathcal{M}, \\ \mathbf{b}_i, & i \notin \mathcal{M}. \end{cases} \tag{3}$$

At training time, the model receives the masked sequence $\tilde{\mathbf{b}}$ and is trained to predict the original binary codes at the masked positions. Unlike index-based token prediction over a $|\mathcal{C}|$-sized codebook, we instead perform bitwise classification. Specifically, let $\mathbf{b}_i = (b_{i,1}, \ldots, b_{i,D}) \in \{-1, +1\}^D$ denote the $D$-dimensional binary code at position $i$. The model outputs probabilities.

$$p_\theta(b_{i,j} \mid \tilde{\mathbf{b}}), \quad j = 1, \ldots, D,$$

interpreted as independent Bernoulli distributions over each bit. The masked prediction loss is then

$$\mathcal{L}_{\text{mask}}(\theta) = \mathbb{E}_{\mathbf{b}} \, \mathbb{E}_{\mathcal{M}} \left[ \sum_{i \in \mathcal{M}} \sum_{j=1}^{D} -\log p_\theta(b_{i,j} \mid \tilde{\mathbf{b}}) \right]. \tag{4}$$

At inference time, we follow the iterative parallel decoding strategy of MaskGIT (Chang et al., 2022). The process begins with all tokens masked and proceeds in a fixed number of steps, where at each step the model predicts the masked positions conditioned on the visible context and progressively unmasks the most confident predictions.

## 3 EXPERIMENTS

This section provides a detailed empirical comparison with established baseline methods, as well as ablation of specific design and training choices.

**Training Datasets.** The image tokenizer in this work is trained on the ImageNet-1K (Russakovsky et al., 2015) and MS-COCO (Dosovitskiy et al., 2020) training sets. We train this tokenizer at a

Table 1: Performance comparison of different image compression methods on ImageNet and COCO datasets. We highlight best performance in **bold**, and the second best with underline. Our upper bound is set to WAN-VAE and is highlighted in blue.

| bpp | Method | Dim | Size | ImageNet | | | | COCO | | | |
|-----|--------|-----|------|--------|------|-------|-------|--------|------|-------|-------|
| | | | | LPIPS↓ | rFID↓ | PSNR↑ | SSIM↑ | LPIPS↓ | rFID↓ | PSNR↑ | SSIM↑ |
| 0.039 | VQGAN Esser et al. (2021) | 256 | 1024 | - | 8.30 | 19.51 | 0.614 | - | 16.95 | 19.08 | 0.613 |
| 0.055 | VQGAN Esser et al. (2021) | 256 | 16384 | - | 4.99 | 20.00 | 0.629 | - | 12.29 | 19.57 | 0.630 |
| 0.039 | MaskGIT Chang et al. (2022) | 256 | 1024 | - | 2.28 | - | - | - | - | - | - |
| 0.219 | VQGAN Esser et al. (2021) | 4 | 256 | - | 1.44 | 22.63 | 0.737 | - | 6.58 | 22.29 | 0.744 |
| 0.219 | VQGAN Esser et al. (2021) | 4 | 16384 | - | 1.19 | 23.38 | 0.762 | - | 5.89 | 23.08 | 0.771 |
| 0.201 | ViT-VQGAN Yu et al. (2021) | 32 | 8192 | - | 1.28 | - | - | - | - | - | - |
| 0.218 | LlamaGen Sun et al. (2024) | 8 | 16384 | - | 0.59 | 24.45 | 0.813 | - | 4.19 | 24.20 | 0.822 |
| 0.070 | Flowmo-Lo Sargent et al. (2025) | 18 | $2^{18}$ | 0.113 | 0.95 | 22.07 | 0.649 | - | - | - | - |
| 0.219 | Flowmo-Hi Sargent et al. (2025) | 14 | $2^{14}$ | 0.073 | 0.56 | 24.93 | 0.785 | - | - | - | - |
| 0.281 | BSQ-18 Zhao et al. (2024) | 18 | $2^{18}$ | 0.076 | 1.14 | 25.36 | 0.759 | 0.074 | 5.81 | 25.08 | 0.766 |
| 0.562 | BSQ-36 Zhao et al. (2024) | 36 | $2^{36}$ | **0.043** | **0.41** | 27.88 | 0.841 | 0.041 | **3.42** | 27.64 | 0.848 |
| 0.218 | **Ours** | 14 | $2^{14}$ | 0.103 | 3.21 | 25.41 | 0.753 | 0.102 | 10.20 | 25.06 | 0.760 |
| 0.500 | **Ours** | 32 | $2^{32}$ | 0.055 | 1.77 | 28.41 | 0.848 | 0.052 | 5.83 | 28.07 | 0.853 |
| 0.562 | **Ours** | 36 | $2^{36}$ | 0.049 | 1.33 | 29.29 | 0.860 | 0.046 | 5.00 | 28.91 | 0.867 |
| 0.562 | **Ours-L** | 36 | $2^{36}$ | 0.048 | 1.40 | **29.52** | **0.863** | **0.041** | 4.98 | **29.24** | **0.871** |
| 4 | WAN-VAE Wan et al. (2025) | 16 | - | 0.032 | 1.03 | 31.42 | 0.889 | 0.027 | 3.49 | 31.24 | 0.896 |
| 1 | SD-VAE Rombach et al. (2022) | 4 | - | 0.098 | 1.35 | 21.99 | 0.627 | 0.099 | 5.94 | 21.68 | 0.638 |
| 1 | SDXL-VAE Podell et al. (2023) | 4 | - | 0.066 | 0.72 | 25.38 | 0.727 | - | 4.07 | 25.76 | 0.845 |

$256 \times 256$ spatial resolution obtained by bilinear interpolation. For videos, we train a dedicated video tokenizer on the PANDA dataset (Chen et al., 2024) to encourage broad generalization, rather than adapting an image tokenizer to video as in prior work (Zhao et al., 2024). For a fair comparison with baselines, we resize all videos to $128 \times 128$ using bilinear interpolation and train and evaluate our video tokenizer at this resolution, reporting evaluation results on UCF-101 (Soomro et al., 2012).

**Implementation and Training.** We follow the implementation of WAN to initialize the pretrained continuous VAE. This includes a CNN based encoder with three $2\times$ spatial, and two $2\times$ temporal downsampling blocks followed by a symmetric decoder upsampling to the original image/video dimensions (Wan et al., 2025). The tokenizer follows a standard ViT-VQGAN (Yu et al., 2021), with the internal quantization bottleneck being replaced by LFQ, following the details provided in (Yu et al., 2023b), and the BSQ implementation (Zhao et al., 2024). Generation is done using a standard bi-directional transformer under MaskGIT modelling assumption, and bitwise classification described in section 2.2. The architecture for generation is a standard Bert model. We run all our experiments on 32 NVIDIA A100 GPUs.

## 3.1 IMAGE AND VIDEO COMPRESSION

Table 1 reports ImageNet-1K and MS-COCO validation performance, comparing our method against state of the art image tokenization baselines. To facilitate a fair comparison we conduct the image compression evaluations on $256 \times 256$ spatial resolution similar to the baselines (Zhao et al., 2024; Sargent et al., 2025). Furthermore evaluations are done under the same number of tokens per image obtained from the $\times 8$ spatial downsampling of WAN-VAE. We keep the ViT patch size at 1 in order to achieve the same number of tokens as our most competitive baselines BSQ(Zhao et al., 2024), and FlowMo (Sargent et al., 2025). This allows us to modify the bits per pixel (bpp), based on the latent dimension. At equivalent bpp our tokenizer attains competitive results across PSNR, MS-SSIM, and LPIPS, with the largest gains in the high-rate regime, where capacity suffices to faithfully reconstruct the WAN-VAE latent space. Importantly, the discrete tokenization closely tracks the reconstruction quality of the continuous VAE upper bound (frozen encoder/decoder without discretization), supporting our claim of discrete tokens with continuous fidelity. By default, we report results with a small model ($\sim$ 28M parameters); we also present a large variant ($\sim$ 300M parameters, "Ours-L") to quantify the effect of capacity. We show the best and second-best results with **bold** and underline, respectively.

In the high-rate setting ($>0.50$ bpp), our models deliver state-of-the-art distortion performance with competitive perceptual quality across both benchmarks. At 0.562 bpp, our larger model (Ours-L) attains 29.52 dB PSNR / 0.863 SSIM on ImageNet and 29.24 dB / 0.871 on COCO, exceeding BSQ-36 by +1.64 dB / +0.022 (ImageNet) and +1.60 dB / +0.023 (COCO). Perceptually (LPIPS), Ours-L

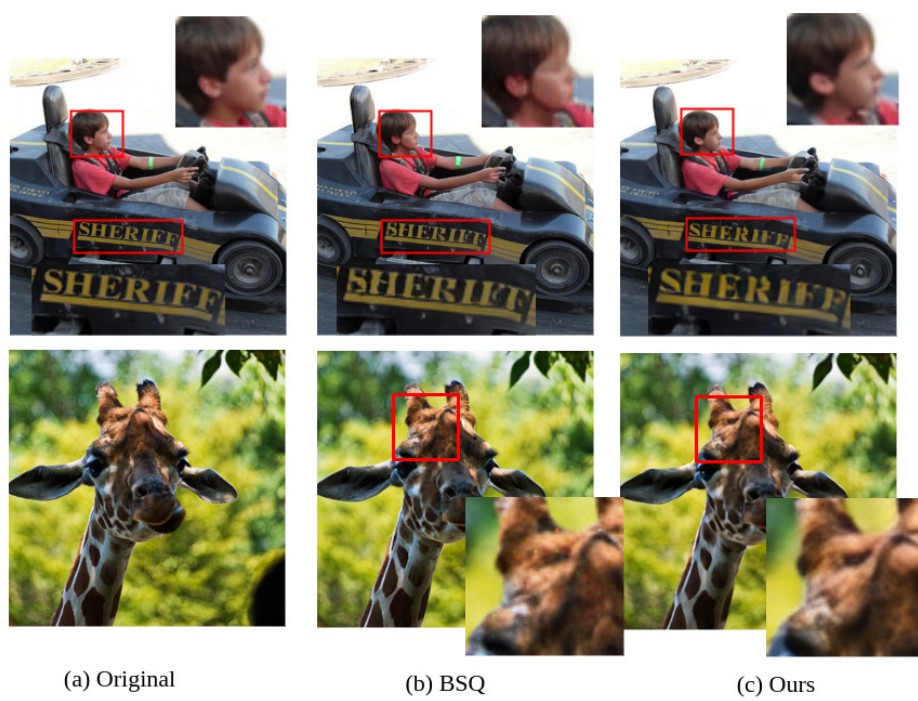

(a) Original                    (b) BSQ                    (c) Ours

Figure 3: Qualitative comparison between our image tokenizer and the competitive baseline BSQ. The bottom row highlights the presence of jitter artifacts in BSQ reconstructions, which are not well captured by FID despite being perceptually evident.

matches BSQ on COCO and is a very close second on ImageNet, while retaining clear advantages in PSNR/SSIM. Notably, even at 0.50 bpp, the 32-dim variant reaches 28.41 dB / 0.848 (ImageNet) and 28.07 dB / 0.853 (COCO), surpassing BSQ-36 on distortion while using fewer bits, indicating that our rate–distortion efficiency persists across models. Our rFID on the other hand, performs weaker than the strongest competitive baseline, while still outperforming most other tokenizers. We trace this to slight, spatially smooth color shifts arising from imperfect recovery of the WAN-VAE latent space due to the small number of available bits.

Table 2 compares reconstruction on UCF-101 (split 1) at $128{\times}128$. In video reconstruction our video tokenizer sets a new state of the art on the fidelity and perceptual similarity among discrete baselines. Compared to prior tokenizers such as MaskGIT, TATS, MAGVIT/MAGVIT-v2, and LARP, our method attains higher PSNR/SSIM with markedly lower LPIPS, yielding visibly crisper frames and fewer compression artifacts. While competitive, our rFVD trails the best by a small margin. Similar to image reconstruction, subtle color drift and softened high-frequency textures from quantization account for most of this difference.

## 3.2    IMAGE GENERATION

We train a bitwise MaskGIT decoder on ImageNet-1K tokens, following the masked modeling and sampling schedule of the reference implementation Chang et al. (2022). MaskGIT is chosen for its parallel decoding and favorable speed–quality trade-off. Our aim here is to assess the tokenization procedure and the viability of bitwise factorization. We evaluate the Fréchet Inception Distance (FID) on ImageNet-1K and compare with recent baselines under their recommended evaluation protocols; our results are reported in Table 3. We show a set of uncurated samples generated by our model in Figure 4.

Table 2: Performance comparison of video compression against other competitive baselines.

| Method | Dim | Size | UCF-101 | | | |
|---|---|---|---|---|---|---|
| | | | LPIPS↓ | PSNR↑ | SSIM↑ | rFVD↓ |
| MaskGIT Chang et al. (2022) | 256 | 1024 | 0.114 | 21.5 | 0.685 | 216 |
| TATS Ge et al. (2022) | 256 | 16384 | - | - | - | 162 |
| MAGVIT-L Yu et al. (2023a) | 256 | 16384 | 0.099 | 22.0 | 0.701 | 25 |
| MAGVIT-v2 Yu et al. (2023a) | 18 | $2^{18}$ | 0.069 | - | - | 16.12 |
| MAGVIT-v2 Yu et al. (2023a) | 18 | $2^{18}$ | 0.054 | - | - | **8.62** |
| LARP-B Wang et al. (2024a) | 8 | 8192 | 0.086 | 27.88 | - | 31 |
| LARP-L Wang et al. (2024a) | 8 | 8192 | - | - | - | 24 |
| **Ours** | 36 | $2^{36}$ | **0.031** | **30.38** | **0.923** | 26.5 |
| WAN Wan et al. (2025) | 4 | - | 0.019 | 35.69 | 0.959 | - |

Table 3: Image generation results on ImageNet-1K (256×256)

| Category | Method | # steps | FID ↓ |
|---|---|---|---|
| Diffusion | ADM | 1,000 | 10.94 |
| | Improved DDPM | - | 12.26 |
| AR | DCTransformer | - | 36.51 |
| VQ-based | VQVAE-2 | - | 31.11 |
| | VQGAN | - | 15.78 |
| | VQGAN* | - | 18.65 |
| Masked LM | VQ | 12 | 9.40 |
| | **Ours** | 32 | 11.2 |

## 3.3 ABLATION

This section provides an ablation study of the key components of the tokenizer and the adopted bitwise MaskGIT strategy. Given the resource-intensive nature of training these models, ablations are reported only for selected datasets, as specified in the caption of the provided tables.

**Tokenizer.** The tokenizer presented comprises of multiple key choices, and objectives. First, we investigate the effect of size of the tokenizer on reconstruction fidelity. We perform the image training at a dimension of 36 on a small model of 50M parameters, and a larger model with 300M parameters. These findings are presented in Table 4a. We find that regardless of the tokenizer size, our model is bounded to the performance of the backbone. In this case, our ablation in 4a demonstrates the superior performance of the smaller models that can be achieved in fewer training steps compared to the larger model. As a result, our final tokenizer follows the small model. Furthermore, we ablate the effect of axillary objectives namely commitment loss, and the entropy loss. The ablated commitment loss follows the standard objective of VQGAN (Esser et al., 2021). Empirically, adding a standard entropy penalty degrades tokenizer performance; by contrast, a commitment loss weighted at 0.25 yields improved perceptual fidelity and rFID. Consequently we opt for a commitment loss for both image and video tokenizers. Furthermore we evaluate the model with both decoders set as learnable. For this setting, we first train the tokenizer following the procedure in Section 2. We then freeze the encoders and fine-tune the decoders for additional steps. As shown in Table 4, joint decoder fine-tuning diminishes the advantage of our framework and further degrades the reconstruction quality.

## 4 RELATED WORKS

Vector-quantized VAEs (VQ-VAE) have long been used as the backbone for visual tokenizers(Van Den Oord et al., 2017). Building on the VQ-VAE, GAN-regularized variant was introduced by Esser et al. (2021) demonstrating the tokenizer as a primary driver of generative quality. Since then,

Table 4: Tokenizer ablation conducted on full COCO validation set.

(a) Decoder joint fine-tuning

| Model | LPIPS ↓ | rFID ↓ | PSNR ↑ |
|---|---|---|---|
| Base | 0.046 | 5.000 | 28.912 |
| Joint finetuned | 0.053 | 6.474 | 28.295 |
| WAN | 0.027 | 3.490 | 31.242 |

(b) Objective effect

| Model | LPIPS ↓ | rFID ↓ | PSNR ↑ |
|---|---|---|---|
| None | 0.054 | 5.580 | 28.422 |
| $\mathcal{L}_e$ | 0.059 | 5.830 | 28.090 |
| $\mathcal{L}_c$ | **0.046** | **5** | **28.912** |

the "vanilla" VQGAN tokenizer has been adopted across autoregressive (AR) and MaskGIT-style decoders, as well as next-scale predictors for images and video generation (Chang et al., 2022; Tian et al., 2024; Yu et al., 2023a). Many works retain this tokenizer while swapping backbones (e.g., CNN→ViT) (Cao et al., 2023; Yu et al., 2021). Yet, irrespective of such architectural changes, vanilla VQGAN exhibits poor codebook utilization and is prone to collapse, ultimately limiting scale.

A concurrent line retains the VQGAN tokenizer and augments training with auxiliary losses or mechanisms to promote codebook usage; Reg-VQ (Zhang et al., 2023), for example, places a prior over discrete codes and mixes stochastic with deterministic quantization to encourage broader utilization. LARP (Wang et al., 2024b) interprets cosine similarities as logits to enable stochastic sampling at commit time. Such stochasticity can raise utilization statistics, but it does not uniformly improve reconstruction and may burden the decoder with noisy assignments. In contrast, VQGAN-LCP (Zhu et al., 2024) removes stochasticity and scales the codebook beyond $99k$ by seeding entries with frozen CLIP (Radford et al., 2021) embeddings. However, subsequent work (Yu et al., 2023b) report diminishing returns as codebooks grow, suggesting a scaling ceiling that is not resolved by increasing vocabulary size alone.

This observation is made explicit by LFQ (Yu et al., 2023b) which shows that merely enlarging the codebook stalls generative quality. LFQ break this barrier by capping per-code representational capacity while scaling the codebook size, effectively disentangling capacity from size. Binary Spherical Quantization (BSQ) Zhao et al. (2024) replaces vector quantization with a projection to a hypersphere followed by binary quantization, yielding a scalable codebook-free tokenizer that scales to arbitrary token dimensions. While their method establishes a compelling compression/reconstruction point, its reliance on spherical binary projections differs from our objective of aligning discrete representations to an underlying continuous metric space, and its AR prior is tailored to entropy coding rather than parallel generative decoding.

On a different line of works, MAR (Li et al., 2024) trains autoregressive transformers over continuous token values by replacing categorical cross-entropy with a diffusion-based per-token loss. Diffusion Tokenizers (DiTo) (Chen et al., 2025) argue that the diffusion objective suffices to learn compact visual representations at scale. These works recasts the decoder as a denoiser or scales the tokenizer to compensate for quantization loss. We take a complementary route and train the tokenizer to naturally recover the VAE's continuous latent space, we can maintain the benefits and simplicity of autoregressive modeling while achieving continuous-grade generation without specialized training procedure, diffusion objectives or scaling the tokenizer.

Driven by the same goal of marrying continuos fidelity with discrete modeling, TokenBridge (Wang et al., 2025) discretizes a frozen VAE post hoc via training-free, per-channel quantization and predicts the resulting indices with a channel-wise autoregressive head. This creates two core bottlenecks: the discretizer is not learned (bins cannot adapt to latent geometry or task semantics), and decoding incurs intra-token sequential dependencies proportional to the VAE's channel count. By contrast, we learn the discrete bottleneck to align with the frozen continuous latent and decode with bitwise MaskGIT in parallel across masked positions and bit-planes, removing channel-wise AR and large K-way heads with complexity linear in bits.

In summary, our approach differs from prior bit-token or binary quantization methods in two ways: (i) we align the discrete bottleneck to a frozen continuous VAE and show that this preserves metric structure and reconstruction quality close to the continuous upper bound, and (ii) we replace large K-way heads with bitwise parallel MaskGIT, which scales output complexity linearly in the number of bits and improves convergence without iterative diffusion refinements. Prior works either rely

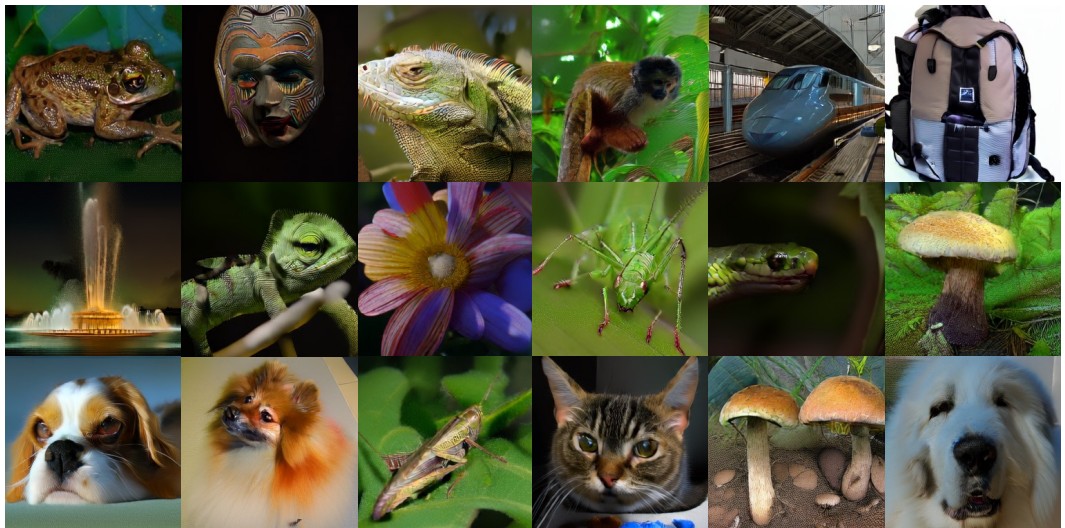

Figure 4: Uncurated results on ImageNet-1k.

on large codebooks, bitwise autoregression, or diffusion-style decoders; we demonstrate a parallel, codebook-free path that remains compatible with language-style training and inference.

## 5 DISCUSSION & CONCLUSION

In this work, we address the long-standing gap between discrete and continuous tokenizers. We introduced a codebook-free discrete tokenizer aligned to a frozen continuous VAE and paired it with a bitwise MaskGIT decoder. This design preserves the practical advantages of discrete modeling, exact token likelihoods, composability, and parallel decoding, while maintaining the metric structure typically associated with continuous latents. By predicting bit-planes rather than a large K-way vocabulary, the decoding head scales with the number of planes instead of the codebook size, reducing parameters and memory, and enabling efficient deployment across images and videos. Our tokenizer tracks a continuous upper bound closely while avoiding axillary GAN losses, or iterative refinement. We demonstrate competitive image reconstruction, and state of the art video reconstruction through

Looking ahead, we see value in multi-resolution extensions that organize bit-planes across spatial scales and in resolution-agnostic training/inference strategies that generalize across sizes without retraining; we also aim to model temporal structure via multi-frame tokens with lightweight conditioners for inter-plane dependencies, and to explore modest backbone adaptation or latent calibration to mitigate residual artifacts and clarify scaling at higher resolutions and frame rates.

**Limitations** Our method inherits the representational capacity of the frozen VAE, which limits our reconstruction capacity to the VAE as the upper bound. Furthermore, Our approach is still fragile at very low bitrates, where small latent mismatches can be amplified by the VAE decoder. Although medium to high bitrate yields alignment, extreme compression remains challenging.

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

## A    APPENDIX

You may include other additional sections here.

