# OpenReview forum: "LatentBit: Discrete Visual Tokenization with Preserved Continuous Structure"
_ICLR.cc/2026/Conference — ICLR 2026 Conference Withdrawn Submission_

### Official Review · Reviewer_JJti · 2025-10-19

**Soundness:** 2
**Presentation:** 2
**Contribution:** 2
**Rating:** 2
**Confidence:** 4

**Summary:**

This paper proposes a discrete visual tokenizer that bridges the gap between discrete and continuous representations for image and video generation.
The key idea is to place a learnable ViT-based tokenizer at the bottleneck of a frozen continuous VAE, converting its latents to binary codes via lookup-free quantization (LFQ).
The authors then introduce Bitwise MaskGIT, which performs token prediction at the bit level rather than predicting full token indices, enabling tractable generation over large vocabularies from the conversion.
The method claims to achieve reconstruction quality approaching continuous VAEs while maintaining the advantages of discrete, language-style frameworks.

**Strengths:**

1. Following previous work, this paper proposed a new method to convert a continuous VAE into a discrete visual tokenizer
2. Experimental results shows strong image reconstruction ability of the proposed tokenizer

**Weaknesses:**

1. the authors claimed better likelihood in the abstract and introduction, however, this is not supported with theory or experimental results.
2. The difference between the proposed bitwise MaskGit and previous work MaskBit is unclear, the authors are encouraged to elucidate this.
3. scalability to large models. For video generation task, all video are in the resolution of 128*128, does this method support higher resolution without bottlenecked by the computation requirement caused by the number of tokens and attention mechanism.

------------------------------------------------------------------------------------------------------------------------------
This method seems to be unfinished
The first part demonstrates appearing reconstruction ability of the proposed discrete tokenizer; however the part shows the generative ability looks a bit in a rush and present result much worse than the current SOTA tokenizers.
Given this limitation, the authors are supposed to train MaskGit and MaskBit baseline for the proposed discrete tokenizer.

**Questions:**

1. could the authors explain "Continuous latent generators (e.g., diffusion in VAE space) achieve high perceptual quality but do not provide tractable token-level likelihoods or truly parallel decoding,", why continuous can not provide true parallel decoding?
2. in line 157-159, how is it still optimizing KL after adding semantic VGG loss

---

### Official Review · Reviewer_5dhe · 2025-10-22

**Soundness:** 2
**Presentation:** 2
**Contribution:** 2
**Rating:** 2
**Confidence:** 3

**Summary:**

This paper proposes predicting bitwise-factorized distributions rather than a “flat” logit vector of the full vocabulary size. The motivation is to enable image generative tasks using autoregressive or masked prediction in the space of LFQ tokenizers with very large discrete vocabularies (codebook sizes of e.g. 2^36). Such very large codebooks enable training discrete versions of continuous tokenizers by simply introducing a few additional bottleneck layers while preserving the (frozen) pretrained encoder and decoder from a continuous tokenizer with minimal loss in reconstruction quality.

**Strengths:**

While the MAGVITv2 paper, which introduces LFQ as a way to efficiently scale up codebook size, already addresses the difficulty of predicting distributions over very large numbers of possible tokens via factorizing their distribution, this paper attempts to push the idea to the extreme by predicting individual bit probabilities. This is an interesting and important direction with the potential to provide empirical understanding of the tradeoffs involved with factorized token distribution modeling.

To this end, the authors first demonstrate that very large codebooks can potentially allow directly adapting continuous tokenizers to large, binary-quantized codebooks with minimal loss to reconstruction quality. The paper shows that this can be accomplished in a simple and efficient manner by learning a few bottleneck layers and using binary quantization, while reusing the frozen encoder and decoder from a continuous tokenizer. The tokenizers trained in this manner approach the performance of the original continuous tokenizer when the codebook size is scaled large enough, and achieve similar performance as tokenizers trained “from scratch” (e.g. BSQ) given a similar bits-per-pixel budget.

**Weaknesses:**

While bitwise-factorized prediction for large-vocabulary masked modeling is the main contribution of this paper, experiments demonstrating its effectiveness for generation are severely limited. In particular, only Table 3 shows any generation results, and the results appear worse than several baselines (although the baselines used are not closely related to the proposed method). Furthermore, the only reported results are FID for ImageNet unconditional generation, which is a less popular benchmark than conditional generation, and is thus missing some important baselines like MaskGIT, MAGVITv2, FSQ and BSQ. There are also some concerns with the baseline results reported (see Q3 below). Overall, the feasibility of using very large discrete vocabularies for image generation is not successfully demonstrated (it seems to me that BSQ with less-than-maximal factorization, as well as FSQ, might perform better), undermining the central motivation for the proposed approach.

Furthermore, while the manuscript (and even the title) mentions “preserving metric structure” of the continuous tokenizer, this property is not precisely defined or otherwise elaborated on. In particular, experiments do not validate its benefits – indeed, reconstruction performance is very similar to BSQ, which is trained with quantization “from scratch”, and generation performance appears to be worse than other VQ or LFQ methods. The paper would be strengthened if experiments and arguments were included to further motivate “bootstrapping” a discretized tokenizer from a continuous one (for example, is overall convergence faster when training a continuous tokenizer first and the switching to discrete tokens later, compared to training with discretization from the beginning — if so, why?).

**Questions:**

**Q1.** You mention that your approach leads to natural codebook utilization (L076). Could you provide codebook utilization metrics to support this claim?

**Q2.** Why was ImageNet unconditional generation chosen as the primary generation benchmark? It would be better to also include class-conditional ImageNet generation, as it is a more popular benchmark and would allow you to compare against e.g. MaskGIT, MAGVITv2, FSQ and BSQ, which are much more closely related and relevant baselines. It would also be helpful to include IS in addition to FID. (In general, I would suggest following the MaskGIT/BSQ/FSQ evaluation protocol as closely as possible, since your method is most related to these approaches.)

**Q3.** In Table 3, the results labelled “VQ” should be properly attributed to the FSQ paper by Mentzer et al. [1] (I am assuming this result comes from their Figure 3, 2nd panel). Furthermore, those FID scores are for ImageNet 128x128 rather than 256x256, so your Table 3 reports inconsistent results and needs to be fixed. Should you instead use the FID reported by FSQ in their Figure 4 [1]?

**Q4.** In order to justify that the bitwise factorization is a sensible choice, I would have liked to see an experiment comparing generation performance as a function of more and more highly factorized predictions. That is, you could start by factorizing the vocabulary into four 9-bit sub-tokens, similar to MAGVITv2, and then sweep across different values (9, 6, 4, 3, 2, 1-bit) until arriving at your proposed bitwise factorization. A result showing that the 1-bit factorization can achieve similar or better results than 2, 3, 4, 6, or 9 bits, would be a crucial and valuable addition to the paper as, without it, I am not convinced of the benefits of a 1-bit factorization compared to less highly factorized alternatives (which also have moderate vocabulary sizes and therefore do not pose any practical problems). Could you consider sharing the results of such an experiment?

**Q5.** In the joint finetuning experiment from Section 3.3, do you use an adversarial loss? Since WAN is fine-tuned with an adversarial loss, I am not surprised that further training without adversarial loss hurts the FID, so if this is the case, the value of the joint finetuning experiment is not clear to me.

---

Minor comments (typos/writing):
* L364 and L464: “axillary” -> “auxiliary”
* L465: unfinished sentence/paragraph
* Unused placeholder appendix should be removed.

---

[1] Finite Scalar Quantization: VQ-VAE Made Simple. Fabian Mentzer, David Minnen, Eirikur Agustsson, Michael Tschannen. ICLR 2024. https://openreview.net/forum?id=8ishA3LxN8

---

### Official Review · Reviewer_gyn6 · 2025-10-28

**Soundness:** 3
**Presentation:** 3
**Contribution:** 3
**Rating:** 4
**Confidence:** 4

**Summary:**

The paper tries to bridge the gap between discrete and continuous tokenizers, and making discrete tokenizers less lossy with the goal of better image/video generation quality. The key idea is relatively simple. It takes a pretrained and frozen VAE (the authors experimented WAN2.1's VAE), adds some transformer layers to serve as a tokenizer with LFP at the bottleneck layers. The intuition is that by doing this the discrete tokenizer directly learn from the continuous counterpart, so that it can preserve the structure and details of the continuous tokenzier.

To leverage the tokenizer, the authors also introduce Bitwise MaskGIT, which extends MaskGIT-style generation to large vocabularies by decomposing token prediction into independent binary bit predictions.

The paper reports reasonable results on image and video reconstruction tasks and image generation results.

**Strengths:**

(+) The design of the tokenizer seems make sense to me and well motivated. The research problem is well motivated. The proposed approach of learning discrete tokens from a frozen continuous latent space makes sense

(+) The writing of the paper is good. I can relatively easily follow the paper.

(+) The reported results seems to be relatively strong

**Weaknesses:**

(-) The proposed method is bounded by the VAE's performance.

(-) Only one VAE (Wan2.1) was experimented and shown results. It is unclear how well and easy this extends to other VAEs

(-) The motivation of the paper is for image and video generations as authors highlight in the intro. However there are very limited image generation results and no video generation results at all. If this is due to limited compute, I suggest the authors to not highlighting the motivations of video generation but focus on image generation tasks and show really good results (and more comprehensive results) on image generation, and leave video generation as a future work

(-) While the intuition of mimicking continuous VAE's latent space seems make sense. Is there any theoretical justification of it?

(-) There are more recent and advanced discrete tokenizers that claim to be close to continuous tokenizers. One example is Infinity (https://github.com/FoundationVision/Infinity), CVPR'25. The comparison methods authors use in this paper seem to be weak.

**Questions:**

1/ How is the token utilization rate in this approach?

2/ There is still a relatively large gap between the discrete tokenizer and Wan2.1 tokenizer, as shown in the tables. Can you analyze why and potential ways to further bridge the gap?

3/ Have the authors considered an ablation of completely freezing the continuous VAE backbone vs not completely freeze the backbone? Maybe after the discrete tokenzier is learned well enough. Consider unfreeze some/all layers to jointly learn together to see if it further improves?

---

### Official Review · Reviewer_rF56 · 2025-10-30

**Soundness:** 1
**Presentation:** 2
**Contribution:** 2
**Rating:** 2
**Confidence:** 4

**Summary:**

This paper proposes a method for learning discrete representations by augmenting a pre-trained VAE with an additional quantization module. This quantization module consists of a ViT based encoder-decoder architecture followed by a binarization quantization layer. The authors claim that this approach effectively preserves the reconstruction fidelity of the original VAE model. Building on this discrete representation, the authors introduce a bit-level MaskGIT model for generative tasks. The efficacy of the proposed method is evaluated through a series of experiments on image reconstruction, image generation, and video generation.

**Strengths:**

- The authors have conducted a set of experiments across multiple domains (image reconstruction, image generation, and video generation) to validate their proposed method.
- The proposed framework is conceptually straightforward and can be trained effectively without relying on adversarial losses (e.g., GANs), which often introduce training instability.

**Weaknesses:**

- The results in Table 1 for the image reconstruction task are concerning. At equivalent bitrates, the proposed model demonstrates significantly inferior performance compared to the BSQ baseline and other established methods. This raises questions about the effectiveness of the proposed quantization scheme in preserving information from the continuous VAE latent space.
- In the image generation experiments (Table 3), the proposed model is built upon a more powerful VAE backbone to derive its discrete tokens. However, paradoxically, its generative performance is weaker than that of prior methods based on traditional VQ-based tokenization. This outcome is counter-intuitive, as one would expect a stronger foundation model to yield better, or at least comparable, results.
- The aforementioned weaknesses in both reconstruction and generation tasks make it difficult to ascertain the practical benefits or superiority of the proposed approach. The core motivation for building a secondary discrete representation on top of an existing continuous VAE representation seems questionable, especially when it leads to a degradation in performance on fundamental tasks. The paper does not sufficiently justify why this two-step process is preferable to more direct methods of learning discrete latents.

**Questions:**

- What is the performance of MAR architecture with the base VAE model used in the paper? If the model operating on the original VAE latents can achieve superior or even comparable results, it would challenge the central motivation for introducing the proposed discrete quantization layer.

---

### Note · Authors · 2025-11-19

**Comment:**

We would like to thank the reviewers for their constructive and detailed feedback; we agree with the raised points and, in order to improve the work accordingly, we are withdrawing our submission.

**Withdrawal Confirmation:**

I have read and agree with the venue's withdrawal policy on behalf of myself and my co-authors.